# Compositional Distributional Semantics with Syntactic Dependencies and Selectional Preferences

**Pablo Gamallo** 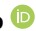

Centro de Investigación en Tecnoloxías da Información (CiTIUS), USC,
15705 Santiago de Compostela, Galicia, Spain; pablo.gamallo@usc.gal

**Abstract:** This article describes a compositional model based on syntactic dependencies which has been designed to build contextualized word vectors, by following linguistic principles related to the concept of selectional preferences. The compositional strategy proposed in the current work has been evaluated on a syntactically controlled and multilingual dataset, and compared with Transformer BERT-like models, such as Sentence BERT, the state-of-the-art in sentence similarity. For this purpose, we created two new test datasets for Portuguese and Spanish on the basis of that defined for the English language, containing expressions with noun-verb-noun transitive constructions. The results we have obtained show that the linguistic-based compositional approach turns out to be competitive with Transformer models.

**Keywords:** compositionality; dependency parsing; meaning construction; compositional distributional semantics; transformer architecture; contextualized word embeddings; sentence BERT

## 1. Introduction

A key element in the process of language comprehension is the ability of humans to combine units of meaning into larger units by means of principles and mechanisms that are not yet well understood. In studies on formal semantics one of the main objectives is to account for the meaning of phrases and compound units from the meaning of their constituent lexical units. In these studies, the Principle of Compositionality, which takes into account the syntactic structure, has been defined to help formalize this semantic process. The Principle of Compositionality defines the meaning of a composite unit as "a function of the meanings of the constituent words and of the way they are syntactically combined" [1].

In the last decade, there have been some works within the distributional semantics approach that have attempted to build distributional semantic models taking into account syntactic structure and applying some version of the Principle of Compositionality to construct the meaning of compound units [2–4]. In these approaches, static word embeddings representing the constituent words of a complex expression or sentence are combined by operations associated with their syntactic categories and syntactic functions within the sentence, so as to elaborate a new embedding that would represent the compositional meaning of the sentence. In some of these syntax-based approaches, the meaning of the sentence is not a single vectorized representation, but the contextualized vectors of each constituent word [3,5]. One of the main challenges is to provide the high-dimensional vector space with composition operations that have clear analogues with operations in formal semantics [6].

More recently, other approaches to contextual distributional semantics, mostly based on Transformers models, have emerged focusing on the same objective: the construction of dynamic and contextual word embeddings. Transformers using pre-training, like BERT [7], have shown impressive results in many natural language tasks, suggesting that these models represent compositional meanings in a very accurate way. However,

because of their black box structure, it is unclear how and to what extent BERT-like Transformers combine word meanings into phrase and sentence meanings to capture high-level compositional information [8]. They do not use any explicit syntactic structure and therefore do not follow the compositional mechanism derived from the Principle of Compositionality. There are some Transformers that incorporate syntactic information in their models, but some doubts arise about the viability of these models in basic Natural Language Processing applications and Information Extraction tasks [9].

The increasing development and use of Transformers and other contextual word models has marginalized compositional distributional semantic models based on syntactic dependencies and compositional operations to meaning construction. One of the main reasons for the lack of interest in these models is their heavy reliance on datasets with simple and controlled syntactic constructions, e.g., noun-verb, noun-verb-noun, adjective-noun, etc. Purely compositional models, due to their linguistic complexity, are so far only successfully applied to datasets with syntactically controlled expressions. In contrast, contextual models derived from Transformers are fully adapted to expressions with open syntax. Therefore, most of the semantic datasets that are made available to the developer community and used in shared tasks contain open text without syntactic restrictions [10,11].

Another important limitation for the development of compositional models is the lack of controlled datasets in different languages. This means that there is no work on multilingual compositional distributional semantics, contrary to what happens in the research focused on Transformers.

In this article, our objective is to design a compositional model for several languages based on syntactic dependencies and to evaluate it on a syntactically controlled and multilingual dataset. For this purpose, we adapt the compositional system described in [12] for a multilingual scenario (English, Portuguese, and Spanish), and created two new test datasets for Portuguese and Spanish on the basis of that defined by Grefenstette et al. (2011) [13] for the English language containing expressions with noun-verb-noun transitive constructions. In addition, the syntax-driven and compositional system will be compared with state-of-the-art Transformers on the mentioned datasets.

The rest of the paper is organized as follows. Related work is introduced in Section 2. Our compositional strategy is defined in Section 3. Material and experiments are described in Section 4. The results obtained in the experiments are shown in Section 5 and discussed in Section 6. Finally, conclusions are addressed in Section 7.

## 2. Related Work

The basic approach to distributional composition, explored by [14], is to just combine vectors of constituent words with arithmetic operations: addition and component-wise multiplication. The main drawback of this approach is that it is not compositional. Word order and syntactic functions are not considered.

Other distributional approaches gave rise to sound compositional models based on Categorial Grammar and functional words represented as high-dimensional tensors [4,15–20]. Our main concern with these approaches is that they require several high-order tensor representations of verbs with n-arity arguments. More precisely, a verb used with a different number of arguments in different contexts should be represented by means of different high-order tensors. Furthermore, the use of high-order tensor results in an information scalability problem as tensor representations grow exponentially as the sentence gets longer.

Our dependency-based approach is inspired by the work described in [21,22], in which selection preferences are formalized in a distributional model. This inspiring work was also the basis for other work with very similar approaches [23,24]. The main difference with regard to our work is that their models were used in a very specific semantic task, namely word sense disambiguation. As they were interested in this specific task, their models were not configured to be used on datasets created for testing semantic composition or sentence similarity.

Another dependency-based and compositional approach is the one reported in [3]. The distributional information of each word is computed from its path in a parse tree. However, they do not make use of the concept of selectional preferences and the strategy tends to build very sparse word representations, especially when a word has a large number of relations in the dependency tree of a sentence.

Finally, in the field of Transformers, very recent approaches make use of syntax information to improve self-attention mechanism [25,26], which results in more compositional semantic strategies.

## 3. A Dependency-Based Compositional Strategy for Meaning Construction

The central idea of the proposed strategy is to give syntactic dependencies a key role in the process of semantic composition. The main proposal of this paper is to put syntactic dependencies at the core of distributional semantic composition. When two words are related by means of a syntactic dependency, two complementary selective functions are activated, each one imposing the selectional preferences of one word on the other one. These two functions allow the two related words to mutually disambiguate or discriminate the sense of each other by co-selection [5]. As distributional word meanings are represented by means of static vectors, the resulting sense of each related word after co-selection is a compositional vector combining the original word vector with the selectional preferences imposed by other word in the dependency. The co-selection process is repeated iteratively throughout the dependency tree with which a composite expression is analyzed. At the end of the process, each word of the compound expression or sentence has a contextualized meaning, as happens when the Transformers models are applied on a text sequence.

### 3.1. Co-Selection

The co-selection process is defined as follows. Given a syntactic dependency linking word $x$, the head, with $y$, the dependent one, by means of syntactic relation *rel* (e.g., obj, nsubj, nmod, etc.):

$$(rel, x, y)$$

two compositional functions are defined, $h$ and $d$, as follows:

$$h_\uparrow(rel, \vec{x}, \vec{y}^{rel\uparrow}) \tag{1}$$

$$d_\downarrow(rel, \vec{x}^{rel\downarrow}, \vec{y}) \tag{2}$$

where $h_\uparrow$ and $d_\downarrow$, respectively, stand for the head and dependent functions; $\vec{x}$ and $\vec{y}$ are the static embeddings respectively associated with the head and dependent words; and $\vec{x}^{rel\downarrow}$ and $\vec{y}^{rel\uparrow}$ are the selectional preferences respectively imposed by words $x$ and $y$. Selectional preferences are vectors dynamically built by considering paradigmatic relations between words [21].

Given the elements of Equations (1) and (2), the head function, $h_\uparrow$, combines (by either multiplying or adding) the static vector of the head word, $\vec{x}$, with the selectional preferences imposed by the dependent one, $\vec{y}^{rel\uparrow}$, so as to construct a new vector representing the contextualized sense of the head: $\vec{x}_{rel\uparrow}$. The dependent function, $d_\downarrow$, combines the static vector of the dependent word, $\vec{y}$, with the selectional preferences of the head, $\vec{x}^{rel\downarrow}$, in order to elaborate a new vector representing the contextualized sense of the dependent: $\vec{y}_{rel\downarrow}$.

Let us see the compound expression "the company fired", which is partially analyzed by linking the head lemma *fire* to the dependent *company*, by means of the nominal subject (*nsubj*) relation (Only lemmas of lexical words are considered. Grammatical words such as auxiliary verbs or determiners are not taken into account so far in our compositional approach.). The application of the two functions defined above in Equations (1) and (2) consists of multiplying (or adding) the static vectors of words with the selectional preferences as follows:

$$h_\uparrow(nsubj, \vec{fire}, company^{nsubj\uparrow}) \quad = \quad \vec{fire} \odot company^{nsubj\uparrow} \quad = \quad \vec{fire}_{nsubj\uparrow} \tag{3}$$

$$d_\downarrow(nsubj, \vec{fire}^{nsubj\downarrow}, \vec{company}) \quad = \quad \vec{company} \odot \vec{fire}^{nsubj\downarrow} \quad = \quad \vec{company}_{nsubj\downarrow} \tag{4}$$

Each function activates a combinatorial operation (component-wise vector multiplication in Equations (3) and (4)), resulting in two new compositional vectors representing the contextualized senses of both the head and the dependent words. In Equations (3) and (4), $\vec{company}^{nsubj\uparrow}$ and $\vec{fire}^{nsubj\downarrow}$ are selectional preferences obtained by adding the vectors of paradigmatically related words:

$$\vec{company}^{nsubj\uparrow} \quad = \quad \sum_{\vec{w}\in\,V} \vec{w} \tag{5}$$

$$\vec{fire}^{nsubj\downarrow} \quad = \quad \sum_{\vec{w}\in\,N} \vec{w} \tag{6}$$

where $V$ is a set of verbal vectors representing the paradigmatic class of those verbs having *company* as subject. This paradigmatic class is used to iteratively build the vector $\vec{company}^{nsubj\uparrow}$, which is obtained by adding the static vectors $\{\vec{w}|\vec{w} \in C_{company/nsubj}\}$ of those verbs (*hire*, *buy*, *invest*, etc) that appear with the noun *company* in the *nsubj* relation. In sum, $\vec{company}^{nsubj\uparrow}$ represents the inverse selectional preferences imposed by the noun *company* on any verb in the subject syntactic position. Notice that in this notation, superscripts are used for specifying selectional preferences vectors while subscripts specify contextualized vectors.

The other vector set, $N$ in Equation (6), stands for the paradigmatic class of nouns appearing in the subject position of *fire*. So, the dynamically constructed vector $\vec{fire}^{nsubj\downarrow}$ is the result of adding the static vectors $\{\vec{w}|\vec{w} \in N_{nsubj/fire}\}$ of those nouns (e.g., *manager*, *director*, *police*, etc.) that could appear as nominal subject of the verb *fire*.

*3.2. Incremental Application of Co-Selection*

Co-selection is the application of two functions on the words related by a specific syntactic dependency. Given the dependency tree associated to a sentence, the sense contextualization of all constituent words is performed by the iterative application of co-selection for each dependency in the tree. So, the composite meaning of a sentence is thus obtained by the recursive process of contextualizing the sense of all its constituent words. As has been said, the sentence is not assigned only one meaning (which could be the contextualized sense of the *root* word), but one contextual sense per constituent lemma. This incremental and iterative process can go either from left-to-right or from right-to-left.

Let us consider now the expression *The company fired an employee*. The semantic interpretation process in the left-to-right direction consists of applying first the co-selection functions of *nsubj* and then those of *obj* (see Figure 1). This results in three contextualized word senses, one per lexical unit: $\vec{company}_{nsubj\downarrow}$, $\vec{fire}_{nsubj\uparrow+obj\uparrow}$ and $\vec{employee}_{nsubj\downarrow+obj\downarrow}$.

Notice that $\vec{company}_{nsubj\downarrow}$ has not been fully contextualized. It is the result of being contextualized by just the verb head, but not by the direct object *employee*. To fully contextualize the nominal subject, the composition process requires to be initialized from right-to-left, as Figure 2 shows.

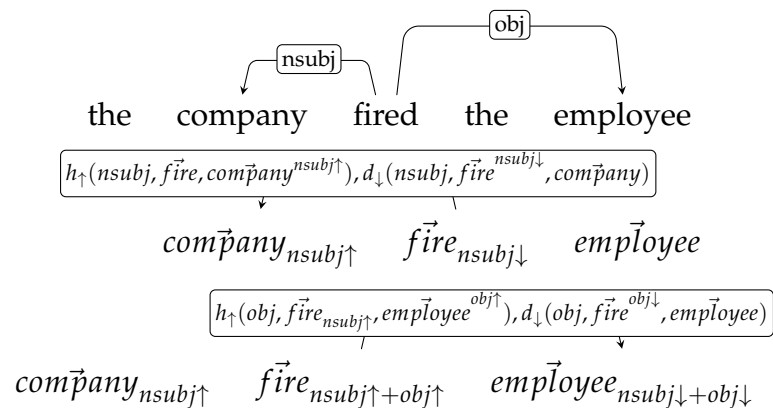

**Figure 1.** Dependency-based analysis of *the company fired the employee* and left-to-right interpretation process to build the contextualized word senses of the three lexical constituents.

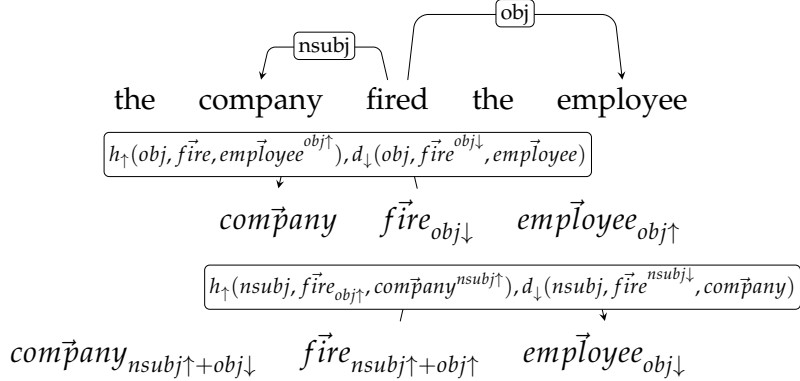

**Figure 2.** Dependency-based analysis of *the company fired the employee* and right-to-left interpretation process to build the contextualized word senses of the three lexical constituents.

## 4. Material and Experiments

In this section, we will compare the performance of the dependency-based compositional approach with models based on Transformers, namely BERT-like models, in a semantic task requiring noun-verb-noun expressions. The experiments will be performed in three different languages: English, Portuguese, and Spanish.

### 4.1. Semantic Task and Test Datasets

To carry out the experiments, we made use of the dataset described in [13], with 199 pairs of transitive expressions and a total of 2487 human judgements on the degree of similarity of each pair. Each pair consists of a transitive verb with both its subject and object nouns (all lemmatized), which are compared to another transitive expression combining the same subject and object with a synonym of the verb that is chosen to be either similar or dissimilar to the verb in the context of the given subject and object. For instance, *report draw attention* is very close to *report attract attention*, as *draw* is a very close synonym of *attract* in that context. By contrast, the same pair of verbs are quite dissimilar in *male draw female/male attract female*. The main problem with this dataset is that, in many cases, there are no degrees of similarity/dissimilarity between expressions, but degrees of compositionality: the second element of the pair is more or less acceptable taking into account compositional (or rather collocational) criteria. For instance, even if *show* and *express* are synonyms in *child show interest/child express interest*, they behave in a very different way in *map show area/map express area*, as in the second one the verb does not fit with the object and the whole expression has low acceptability since it is not a commonly used collocation or word combination. This is a very different case from the one shown previously, where there were two acceptable expressions which were different due to a

change of meaning of the verb. As they are very different cases, some annotators considered the change of meaning as a strong differentiating element, while others considered the differentiating element to be the presence of an expression with low acceptability. This lack of criteria in scoring was compensated and minimized by the fact that the score of each example is the average of up to 13 different annotators rating from 1 to 7.

As there are no datasets with noun-verb-noun expressions for other languages than English, we created two new datasets in Portuguese (193 pairs of transitive expressions) and Spanish (also 193 pairs). The process of developing the two new datasets was as follows. The 199 pairs of the original dataset described in [13] were translated and adapted to Portuguese and Spanish. The author of this article, whose native languages are Galician-Portuguese and Spanish, was responsible for the translation and adaptation of the English dataset to these two languages. He rated each example on a range of 1 to 70, by considering the mean value of the original English example (if any), together with more consistent annotation criteria, so as to minimize the problems previously mentioned. Namely, one of the annotation criteria is to attribute lower similarity values to cases of low acceptability than to cases of change of meaning, although it is always necessary to take into account the gradual factor.

In addition, in order to provide a fair comparison with the contextual models provided by Transformers, we converted the lemma triples of each example into full grammatical sentences in the three languages. For instance the noun-verb-noun triple *map show area* was transformed into the sentence *the map shows an area*. Finally, 6 datasets are available: 3 consisting of lemma triples (one per language) and 3 with full sentences (All datasets are available at https://github.com/gamallo/DepFunc/tree/main/tests (accessed on 15 June 2021)).

In order to evaluate the similarity scores given by all systems, Spearman correlation is computed between individual human similarity scores and the predictions returned by the systems.

### 4.2. System's Configuration

#### 4.2.1. DepFunc

Considering the dependency-based compositional strategy defined above in Section 3, the system called *DepFunc* was updated and set up for the multilingual semantic task introduced in the previous subsection (The system is available at https://github.com/gamallo/DepFunc (accessed on 15 June 2021)). The first version of DepFunc was fully described in [12]. Although the dependency-based strategy is designed to work with any type of phrase and sentence, in practice our system only adapts to linguistic constructions with a fixed and predefined syntactic structure.

DepFunc builds static and count-based word vectors by computing word occurrences in lexico-syntactic contexts. This is performed by making use of the multilingual dependency parser DepPattern [27] and the count-based strategy described in [28], where only the most relevant lexico-syntactic contexts are selected. With DepFunc it is also possible to construct vectors by inverting what is considered object and context, resulting in vectors of lexico-syntactic units (e.g., *company* in the subject position), whose dimensions are the words (seen as contexts) that appear in those syntactic positions.

In a recent work [29], we used as static vectors word embeddings built with *word2vec*, configured with CBOW algorithm, window of 5 tokens, negative-sampling parameter of 15, and 300 dimensions [30]. However, the final compositional vectors derived from these word embeddings yielded worse results on several semantic tasks than syntax-based count-based vectors. This is why we decided to use the latter vectors with DepFunc in the current experiments. Static vectors were built from the English, Spanish, and Portuguese wikipedias (dump files of November 2019 (https://dumps.wikimedia.org/enwiki/) (accessed on 15 June 2021)), containing over 2500 M, 1000 M, and 700 M words, respectively.

Several configurations of the dependency-based compositional approach have been implemented in DepFunc. First, we distinguish between the distributional models of lemmas from those based on lexico-syntactic contexts:

**lemma**: vectors are made of lemmas (objects) in lexico-syntactic positions (dimensions)

**lemma + rel**: vectors are made of lexico-syntactic positions (objects) occurring with lemmas (dimensions)

Then, considering the direction of the compositional process, we can build several compositional vectors representing the meaning of the noun-verb-noun expression. Given the expression *the company fired the employee*, the following compositional vectors representing different points of view of the composite expression can be built:

**left-to-right head**: This builds the compositional vector of the verb head *fire*, as a result of being contextualized first by the selectional preferences imposed by the nominal subject *company* and then by the selectional preferences of the direct object *employee*.

**left-to-right dep-obj**: This builds the compositional vector of the direct object *employee*, as a result of being contextualized by the preferences imposed by *fire* previously combined with the subject *company*.

**left-to-right sentence**: The addition of the two previous left-to-right values (head and dep).

**right-to-left head**: This builds the compositional vector of the verb head *fire*, as a result of being contextualized first by the selectional preferences imposed by the direct object *employee* and then by the selectional preferences of the subject *company*.

**right-to-left dep-subj**: This builds the compositional vector of the subject *company*, as a result of being contextualized by the preferences imposed by *fire* previously combined with the direct object *employee*.

**right-to-left sentence**: The addition of the two previous right-to-left values (head and dep).

Notice that In the left-to-right direction the object (noted dep-obj) is fully contextualized by the verb and the subject; by contrast, the subject is not contextualized by the object, so that this partially contextualized sense of the subject is not used to represent the sentence in any configuration. In the right-to-left direction the only dependent word that is fully contextualized is the subject (dep-subj).

Component-wise multiplication was selected as the compositional function combining words with selection restrictions.

### 4.2.2. BERT-Like Transformers

Concerning the Transformers, for English, we made use of Sentence-BERT (SBERT), a modification of the pre-trained BERT-Large (with 24 layers), to derive fixed sized vectors of sentences [31]. It is very important to note that SBERT for English was fine-tuned with two very large dataset collections, both SNLI [32] and MultiNLI [33] containing 1 million sentence pairs which were annotated for semantic tasks such as inference, contradiction and entailment. In addition, to generate the dynamic vectors of words in context, we used BERT-Large model by adding the 4 last layers of each word. These specific configurations turned out to give the best performances in the experiments reported in [29].

For Spanish and Portuguese, we used BETO [34], with a size similar to BERT-Base (12 layers), and BERTimbau Large [35], with 24 layers, respectively. They are both accessible from HuggingFace Models Hub (https://huggingface.co/dccuchile/bert-base-spanish-wwm-cased (accessed on 15 June 2021) and https://huggingface.co/neuralmind/bert-large-portuguese-cased (accessed on 15 June 2021)). To derive fixed sized sentence embedding, we compute the mean of all output vectors. This is the same strategy as that used by SBERT for English. The main difference with regard to SBERT is that the pre-trained models are not fine-tuned with annotated collections of semantically similar pairs of sentences. As previously shown for English, contextualized vectors were generated by adding the 4 last layers of each word in context.

In sum, four different BERT configurations were used in the experiments:

**SBERT-sentence**: Sentence vector derived from the mean of all output vectors and fine-tuned on NLI and STS datasets.

**bert-sentence**: Sentence vector derived in the same way but without fine-tuning.

**bert-head**: Contextualized vector of the verb head (adding 4 last layers)

**bert-dep-subj**: Contextualized vector of the subject word (adding 4 last layers).

**bert-dep-obj**: Contextualized vector of the direct object (adding 4 last layers).

We consider that any of the three words of the noun-verb-noun expression can semantically represents the whole sentence, as all constituent words are fully contextualized in a BERT-like Transformer architecture.

All systems were evaluated against three datasets (one per language) with examples codified in subject-verb-object triples. However, as we have said before, in order to provide a more appropriate scenario for Transformers, lemma triples were converted into full grammatical sentences. So, for each BERT configuration we defined two different tests:

**Lemma triple**: this corresponds to the original subject-verb-object triple.

**Full sentence**: each triple was transformed into a regular inflected sentence by using the following procedure: by default, a definite determiner is added to the subject, the verb is written in the present tense and the object is preceded by an indefinite determiner. The cases in which this by default procedure does not work have been revised and corrected.

## 5. Results

All systems were evaluated using the three datasets (one per language) described in the previous section. Table 1 shows the results of several configurations of the dependency-based compositional strategy implemented with DepFunc, all of them introduced above in Section 4.2. The left side shows the Pearson scores with lemmas in English, Portuguese and Spanish, while the right side shows the same experiments with lexico-syntactic units. In the first row, the table also shows a non-compositional baseline strategy just comparing head vectors.

The best configurations for Portuguese and Spanish are the right-to-left lemma-based strategies (*lemma_right-to-left*). In Portuguese, the addition of head and dependent (called *sentence*) reaches 45 Pearson correlation, while in Spanish, the best score is achieved with just the contextualized head: 55 Pearson correlation. In English, the best configuration is also achieved with a right-to-left strategy, 53, just considering the contextualized head but using lexico-syntactic units instead of lemmas. To the best of our knowledge, this value is very close to the highest score, 54, obtained by a compositional system on the English dataset [36], and outperforms other compositional methods whose values for the English dataset are also shown in Table 1 (last rows in left side), namely [37,38], and the neural network method reported in [39].

As far as the BERT configurations is concerned, the left side of Table 2 depicts the Speaman scores with lemma triples whereas the right side shows the scores with full sentences. As expected, the use of full sentences consistently improves results in all three languages and in almost all configurations. The only exception is SBERT in English, which achieves its best result with lemmas, 61. This score is in fact the highest value reported so far on this dataset.

**Table 1.** Spearman correlation between different configurations of DepFunc and human judgments on subject-verb-object expressions. Correlation was computed on benchmarks for English (en), Portuguese (pt) and Spanish (es). Top: scores with configurations based on vectors of lemmas. Bottom: scores with configurations based on lexico-syntactic units (lemma + rel).

| Models | $\rho$ (pt) | $\rho$ (es) | $\rho$ (en) |
|---|---|---|---|
| nocomp_lemma—head | 28 | 29 | 29 |
| lemma_left-to-right—sentence | 42 | 29 | 33 |
| lemma_left-to-right—head | 43 | 21 | 35 |
| lemma_left-to-right—dep-obj | 29 | 26 | 19 |
| lemma_right-to-left—sentence | **45** | 50 | 43 |
| lemma_right-to-left—head | 43 | **55** | 35 |
| lemma_right-to-left—dep-subj | 35 | 35 | 44 |
| Grefenstette and Sadrzadeh [37] | - | - | 28 |
| Hashimoto and Tsuruoka [39] | - | - | 42 |
| Polajnar et al. [38] | - | - | 33 |
| Wijnholds et al. [36] | - | - | 54 |
| **Models** | $\rho$ **(pt)** | $\rho$ **(es)** | $\rho$ **(en)** |
| lemma+rel_left-to-right—sentence | −6 | 30 | 19 |
| lemma+rel_left-to-right—head | −1 | 40 | 18 |
| lemma+rel_left-to-right—dep-obj | 2 | 4 | 10 |
| lemma+rel_right-to-left—sentence | 34 | 32 | 27 |
| lemma+rel_right-to-left—head | 38 | 39 | **53** |
| lemma+rel_right-to-left—dep-subj | 1 | 3 | −13 |

**Table 2.** Spearman correlation between different configurations of BERT and human judgments. Correlation was computed on benchmarks for English, Portuguese and Spanish. Top: scores obtained using lemma triples (original dataset). Bottom: scores with full sentences derived from the triples.

| Models | $\rho$ (pt) | $\rho$ (es) | $\rho$ (en) |
|---|---|---|---|
| lemma_SBERT—sentence (fine-tuned) | - | - | **61** |
| lemma_bert—sentence | 32 | 19 | 10 |
| lemma_bert—head | 27 | 34 | 38 |
| lemma_bert—dep-subj | 16 | 27 | 9 |
| lemma_bert—dep-obj | 27 | 32 | 21 |
| **Models** | $\rho$ **(pt)** | $\rho$ **(es)** | $\rho$ **(en)** |
| full_SBERT—sentence (fine-tuned) | - | - | 55 |
| full_bert—sentence | 41 | 28 | 30 |
| full_bert—head | 43 | 42 | 45 |
| full_bert—dep-subj | 33 | 30 | 27 |
| full_bert—dep-obj | 39 | 31 | 36 |

## 6. Discussion

The analysis of the results presented in the previous section leads us to draw some conclusions about the strategies compared in the experiments.

Concerning the compositional method, lemmas are more consistent than lexico-syntactic units. The latter have very irregular behavior when constructing meaning from left-to-right and when using the dependent unit as a representative of the sentence. Only heads from right-to-left work properly.

Still in the compositional model, the left-to-right process seems to build less reliable compositional vectors than the right-to-left counterpart, at least in this specific dataset. This might be due to the weak semantic motivation of the selectional preferences involved in the subject dependency of transitive constructions in comparison to the direct object.

With respect to the results of the Transformers with lemmas, except SBERT (configured only for English), the rest of the systems generally perform worse than the different configurations of the compositional approach with lemmas. To the best of our knowledge, the score achieved by SBERT is, by far, the highest correlation value reported on this

dataset. Our results coincide exactly with those reported in [36] for this system and this dataset. It should be noted, however, that SBERT is a supervised approach as it relies on more than 1 million annotated sentence pairs. Concerning the results of the Transformers with full sentences, they achieve similar scores to those reached by the lemma_right-to-left compositional strategy, except for Spanish, where the compositional method clearly outperforms the BERT scores.

Figure 3 shows the best score of each model per language. Even though the supervised configuration with lemmas, lemma_SBERT, clearly outperforms all models in English, the two more consistent models across the three languages seem to be both the lemma-based compositional model (called DepFunc_lemma on the plot) and SBERT with full sentences (full_SBERT). The former achieves the highest scores for Portuguese and Spanish, while the latter doubles the values obtained by BERT with lemmas and drops slightly with SBERT. that going from lemmas to inflected forms has the least impact on English since it is the least morphologically rich language.

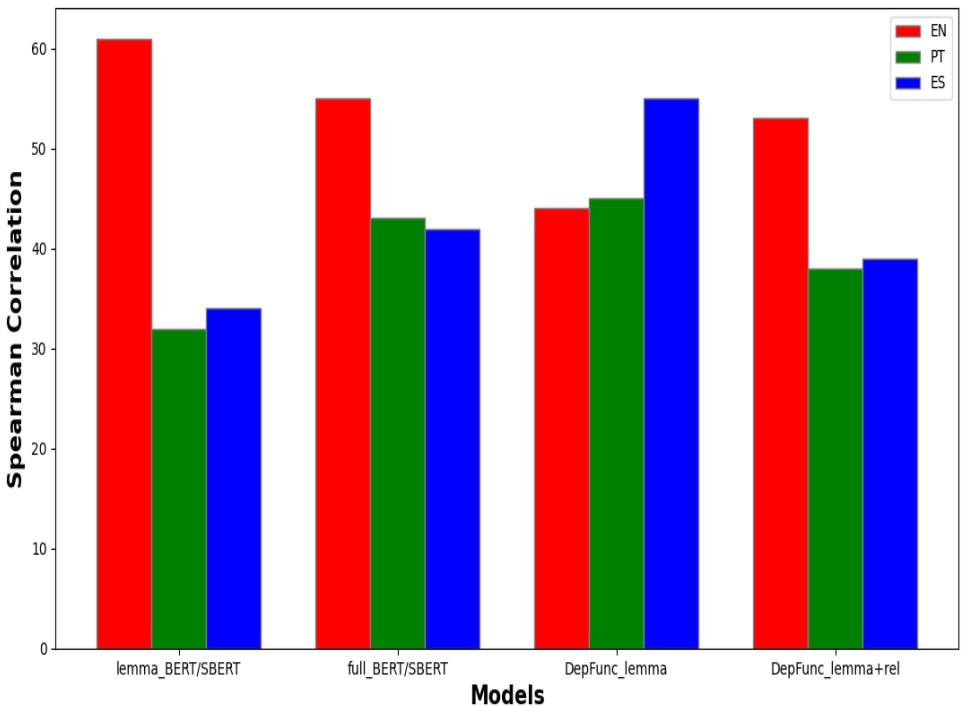

**Figure 3.** Bar plot with the Spearman scores of the best configurations for each model and language.

Finally, we must list some inconsistencies for which we do not have clear answers. On the one hand, the lemma_right-to-left model works significantly better than lemma_left-to-right for Spanish, but not for Portuguese. On the other hand, lemma+rel_left-to-right completely breaks down for Portuguese, but not for Spanish. These discrepancies cannot be explained on the basis of internal differences between the two languages, as they are very close both lexically and syntactically. These divergences could be due to issues of the test dataset and/or problems of the syntactic parsing used to construct the vectors. For instance, some words in the test dataset could be sparse and infrequent in the required syntactic positions within the analyzed corpora of one language, but not of the other. A thorough error analysis of the results will be necessary to reach any conclusions.

Another inconsistency is that SBERT, the supervised model, works better with lemmas than with sentences, unlike the unsupervised BERT configurations in all three languages. To find out if this was the case with other SBERT internal models, we compared several ones: BERT-large (the model we used in the previous experiments with SBERT), BERT-base, DistilBERT-base, and RoBERTa-base. The results obtained with different SBERT models, shown in Table 3, are not conclusive. While the lemmas perform better with the classical

BERT models (base and large), the same is not true for the distilled ones, DistilBERT and RoBERTa, which perform better with full sentences. However, the differences between lemmas and sentences with SBERT is not as marked as in the case of the unsupervised models, as shown above in Table 2, where the use of full sentences leads to results that are twice as good as the use of lemmas. In Table 3, by contrast, the differences between lemmas and full sentences are very small. One possible explanation for this behaviour (suggested by a reviewer) is that going from lemmas to inflected forms has the least impact on English since it is the least morphologically rich language.

**Table 3.** Spearman correlation between different models of SBERT and human judgments for English. Top: scores obtained using lemma triples (original dataset). Bottom: scores with full sentences derived from the triples.

| Models | $\rho$ (en) |
| --- | --- |
| lemma_SBERT—BERT-large | 61 |
| lemma_SBERT—BERT-base | 51 |
| lemma_SBERT—DistilBERT-base | 38 |
| lemma_SBERT—RoBERTa-base | 47 |
| **Models** | $\rho$ **(en)** |
| full_SBERT—BERT-large | 55 |
| full_SBERT—BERT-base | 50 |
| full_SBERT—DistilBERT-base | 41 |
| full_SBERT—RoBERTa-base | 53 |

## 7. Conclusions

In this article, we described a fully compositional strategy based on static distributional models, syntactic dependencies and selectional preferences modeled as paradigmatic relations, which turned out to be competitive when compared to several configurations of BERT in three languages. The fact that compositional models are competitive is a very important factor as they use far fewer parameters in model training than Transformers.

It is important to note that the proposed distributional model is not based on neural networks and results in transparent vectors, which will facilitate explainability in future work. Transparency enables explainability because it helps to track and select the most relevant linguistic features within the compositional vectors.

It is also worth noticing that the proposed compositional strategy is not based on Categorial Grammar and typed functional application as other similar works on distributional compositional semantics [15,20]. In our model, there are no function words like verbs or adjectives, represented as n-order tensors applied on nominal vectors. Function application is driven by binary syntactic dependencies in a semantic space without syntactic constituents. The semantic space proposed in this paper is actually mapped to the Dependency Grammar structure [5].

However, our method has several weaknesses. Among them, it is too dependent on syntactic parsing and, therefore, it has a vulnerable exposure to parser errors. In addition, another notable weakness is the difficulty in processing open text with all kinds of syntactic constructions.

In future work, we will study the possibility of designing and developing a model with fully contextualized vectors for open sentences with any syntactic structure. This will be done by defining a more generic concept of selectional preferences, applicable to any dependency tree in an iterative and incremental process of co-selection. For this purpose, we will also consider the compositional meaning of determiners [40], auxiliary verbs, or tense affixes, which are quantificational operators with different compositional behaviour as lexical units.

**Funding:** This work has received financial support from DOMINO project (PGC2018-102041-B-I00, MCIU/AEI/FEDER, UE), eRisk project (RTI2018-093336-B-C21), the Consellería de Cultura, Educación e Ordenación Universitaria (accreditation 2016-2019, ED431G/08, Groups of Reference: ED431C 2020/21, and ERDF 2014-2020: Call ED431G 2019/04) and the European Regional Development Fund (ERDF).

**Data Availability Statement:** Software and datasets are available in a publicly accessible repository: https://github.com/gamallo/DepFunc (accessed on 15 June 2021).

**Conflicts of Interest:** The authors declare no conflict of interest. The funders had no role in the design of the study; in the collection, analyses, or interpretation of data; in the writing of the manuscript, or in the decision to publish the results.

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
