# Peer review of "Compositional Distributional Semantics with Syntactic Dependencies and Selectional Preferences"

_applsci, doi:10.3390/app11125743_

Round 1
Reviewer 1 Report
This paper proposes a new way to build contextualized word embeddings that is based on syntactic dependencies. The static word embeddings for a given word are modulated by the embeddings of its dependents and its head. The embeddings are tested on a "sentence similarity" task for three languages and compared to related work and to an off-the-shelf BERT model. I find the proposed approach quite appealing and would like to see it published in some form, but - unless I have fundamentally misunderstood something - I do not trust the chosen evaluation. In consequence, the results could not convince me hat this approach is competitive with the cited alternatives.
- Is it correct that you're feeding the lemma triples as input to all your models? If so, this puts BERT at a disadvantage (because it has been trained on full sentences with determiners, punctuation signs and inflected word forms) and your model at an advantage (because you have gold dependencies where the first word of the triple is always the subject of the second and the third word is always the object of the second). In order to provide a fair comparison, I would at least suggest to transform the lemma triples to full grammatical sentences.
- Table 1 shows that lemma_right-to-left works significantly better than lemma-left-to-right for Spanish, but not for Portuguese. On the other hand, lemma+rel_left-to-right completely breaks down for Portuguese, but not for Spanish. Why is that so? I cannot imagine any fundamental syntactic properties between the two languages that should account for these differences, and the effect is not really discussed in the text. This raises the question how reliable in general the results are. How much variation do you get by using different static embeddings as a starting point, or by doing some bootstrap resampling of the test data?
- If I understand correctly, an evaluation instance consists of two lemma triples with the same subject and object, but different verbs. In that case, does it make any sense to evaluate sentence similarity based on the dependent embeddings (all the dep scores in Tables 1 and 2)? If I give the two sentences "map show area" and "map express area" to a model and it produces completely identical contextualized embeddings for the two occurrences of "area", isn't this totally fine? (It may not happen in practice, but I don't see a theoretical reason for evaluating the dep conditions.)
- If the proposed model is useful for sentence similarity tasks, it would also be relevant to evaluate it on a different dataset/task. What comes to my mind is a paraphrase detection task, for which multilingual datasets (PPDB, OpusParcus...) are available.
- I find section 1 quite hard to read, and figures 1 and 2 are not commented in sufficient detail to help me understand the actual model. Sometimes superscripts are used and sometimes subscripts, without indicating their differences. In connection with this, I fail to see the intuition behind distinguishing left-to-right and right-to-left order. If you build the contextualized embedding for the verb, why does the order of contextualization matter? It's all just products and sums, no? Since you're using dependency trees, I would find it more intuitive to distinguish a top-down (from root to leaves) and a bottom-up (from leaves to root) order. Would this be possible in your framework?
- If the Spanish and Portuguese datasets are not presented elsewhere, some more details about the annotation process would be important: How were the triples defined and paired? How are the similarity scores annotated? How many annotators were there, and what was the inter-annotator agreement? Do the 193 Spanish pairs mean the same as the 193 Portuguese or are they different?
Minor comments and typos:
- The three superscript symbols after the author name do not correspond to the four legend symbols before the abstract.
- L7: Grefenstette
- §0 (and throughout the paper): You're aiming at the wrong target when you refer to "Transformers" but really mean "contextualized embeddings from pre-trained language models". For example, everything you say about "Transformers" also applies to ELMo, which is not based on the Transformer architecture but on LSTMs instead. And Transformers are most typically used for machine translation, which is entirely different from the task you're trying to address.
- L40: *to* what extent
- Line numbers are missing after L89 and after L101. Not sure how relevant this is for publication.
- Can equations (3) and (4) be written in a larger font size?
- Footnote 1: considered
- Footnote 1: In the light of your evaluation task, it totally makes sense to only consider lexical words. But are there any fundamental obstacles (regarding computational complexity, for example) against using all words of a sentence?
- L152: there *were* two
- L187: implemented ... implemented
- L336: whatever => any
Author Response
Please, find the attachment

Reviewer 2 Report
The paper proposes an interesting work on a compositional model based on syntactic dependencies for distributional semantic composition. The work is somewhat well presented, and experiments have been done on three different datasets based on three different languages.
However, there are a few concerns that may need to be addressed:
- The various sections are numbered starting from 0. It would be better to start from 1 for Introduction, and so on. Related work is mentioned as section 5. It would be better to place it after Introduction, as section 2.
- In the second paragraph of Results section, the score by SBERT for English is shown as 60. Isn’t this 61 according to Table 2?
- Table captions should appear above the tables.
- In Table 1, it would be good to include the reference numbers also for the last 4 entries (i.e. [25], [26], [27], [24]). For the Polajnar et al. (2015) in Table 1, isn’t the result 43? For Hashimoto and Tsuruoka (2014) in Table 1, the year is 2015 and there is no result that says 43 in the paper. Is it 35?
- A plot showing the test results of the different configurations for the three datasets would be good.
- Line numbers must be removed.
Reviewer 3 Report
This paper focuses on design a compositional model for languages based on syntactic dependencies. Their motivation is based on challenges in providing the high-dimensional vector space with composition operations that have clear analogs with operations in formal semantics. They try to overcome the lack of controlled datasets in different languages in compositional models. The structure of this paper is clear, sections are readable and understandable. However, it should still go through careful revision by considering the following comments.
- The notation of the equations should be better by using short annotate (e.g expression V_{company/nsubj} is too long in eq.5).
- It could be interesting if the author makes an evaluation between their method with the existing method (word2vec CBOW algorithm with 300 dimensions)
- Since English, Portuguese, and Spanish are based on Latin characters, I wonder whether this model can be applied to other languages with hieroglyphs such as Korean, Chinese, etc.
- Some minor mistakes in the English presentation should be corrected.
Round 2
Reviewer 1 Report
Thanks for the updated version and the author response. Unfortunately, the updated paper PDF got jumbled a lot, such that I can barely even read it. Therefore, my comments here are mostly based on the author response.
The additional experiments on BERT with full sentences are really interesting and instructive. I would imagine that going from lemmas to inflected forms has the least impact on English since it is the least morphologically rich language. This might be a partial explanation of the SBERT inconsistency...
Regarding the contextualization of dependent embeddings, I'm still not quite convinced... You say that 'in addition, the word "area" in both "map show area" and "map express area" is not the same even if it is very similar'. This may or may not be true, but I don't think there is a theoretical foundation for your claim. What if you compare "The map shows the area" with "The map showed the area" or "The map will show the area"? In any case, this is a minor issue, and it probably doesn't hurt to show the numbers in the paper anyway...
Regarding my question about the order of contextualization: I think the question is whether it makes sense to determine the order linearly (right-to-left / left-to-right) or based on the dependency labels (subj-before-obj / obj-before-subj). In your dataset, you cannot distinguish the two options, I suppose, but this could be relevant if you want to generalize your work to other syntactic constructions and to languages with different word orders...
I would have found it helpful if some further steps were made to explain the discrepancies between the Spanish and Portuguese results. However, the other reviewers seem to be satisfied with the current state of the paper, so I will not dwell on this any further.
I am choosing "minor revision" because the PDF in its current form is jumbled and text and formulas are not correctly displayed. If these issues are fixed (and the other reviewers agree), I can accept the paper.
